# Diversity of Shallow-Water Species in Prawn Trawling: A Case Study of Malindi–Ungwana Bay, Kenya

Esther N. Fondo [1,*], Johnstone O. Omukoto [1,2], Nina Wambiji [1], Gladys M. Okemwa [1], Pascal Thoya [1], George W. Maina [3] and Edward N. Kimani [1,*]

1  Kenya Marine and Fisheries Research Institute, Mombasa P.O. Box 81651-80100, Kenya; jomukoto@gmail.com (J.O.O.); nwambiji@gmail.com (N.W.); gladysokemwa@gmail.com (G.M.O.); pascalthoya@gmail.com (P.T.)
2  Lancaster Environment Centre, Lancaster University, Lancaster LA1 4YQ, UK
3  The Nature Conservancy, Africa Regional Office, Nairobi P.O. Box 19738-00100, Kenya; gwmaina@tnc.org
*  Correspondence: efondo@yahoo.com (E.N.F.); edwardndirui@yahoo.com (E.N.K.)

**Abstract:** Bottom trawling is a common fishing method that targets bottom-dwelling fisheries resources. It is non-selective and large amounts of by-catch are discarded, raising serious sustainability and ecosystem conservation concerns. In this study, a shallow-water bottom-trawl fishery was evaluated using logbook catch data between 2011 and 2019 and the species composition data collected by fisheries observers between 2016 and 2019. The logbook data showed a twenty-fold increase in the annual catches with a ten-fold increase in fishing effort and an increase in the proportion of retained catch from 2011 to 2019. The observer data showed that for prawn, the by-catch ratio ranged from 1:3 to 1:9 during the four years. Multivariate analysis revealed significant differences between the compositions of retained and discarded catches mainly attributed to *Pellona ditchela*, *Nematopalaemon tenuipes,* and *Secutor insidiator*. There was no significant decline in species diversity and the trophic level of the catches over the 4-year observer period indicating no marked impact of trawling on the stock at the current level of fishing effort. This study provides baseline information on the prawn trawl fishery against which the performance of the management regulations may be evaluated towards the Ecosystem Approach to Fisheries management.

**Keywords:** shell-fish; fish; by-catch; discards; species composition

## 1. Introduction

Bottom trawling and dredging contribute a significant part to capturing finfish and shellfish worldwide. Global evaluation of the contribution of bottom trawling and dredging to capture fisheries indicate approximately 28% [1], while long-term FAO data show that trawling has contributed about 25% of capture fisheries between 1950 and 2018 [2]. However, evaluation shows that bottom trawling generates the most waste in fisheries, accounting for nearly 60% of the fish dumped back into the ocean [1]. The sustainability and conservation concerns of bottom trawl fisheries have attracted attention in the past [1,3–5]. Trawling is considered a wasteful and destructive fishing method associated with large amounts of discarded by-catch leading to changes in trophic structure and loss of fishery resources [6,7].

Changes in the trophic structure and function of benthic communities have important implications on primary production and the wider functioning of the marine ecosystem [8,9]. Ecological studies on bottom trawling have focused on ecosystem impacts through widespread physical disturbance of the bottom substrate, excessive removal of juveniles, and the potential of modifying ecosystems' trophic structure [10,11]. There are claims that any trawl fishery is unsustainable due to its environmental and ecosystem impacts [12,13], and there have been suggestions for bans on some types of trawling [3,14,15]. However, bottom trawling continues to be one of the most common fishing methods and

contribute a significant part of demersal fish and shallow and deep-water crustaceans in many parts of the world's oceans [1].

Bottom trawling within the Western Indian Ocean contributes significantly to industrial shallow-water prawn fisheries in South Africa, Mozambique, Tanzania, Kenya, and Madagascar. The few recent reports on bottom trawling in the region indicate sustainability and conservation challenges. Prawn trawling around Bagamoyo/Sadani and the Rufiji Delta in southern Tanzania, landed between 400 and 1000 tons by 16–26 vessels annually [16–18]. In Mozambique, trawling at the Sofala Bank region yielded 6000 to 9000 tons annually, landed by between 50 and 90 vessels between 1980 and 2014 [19]. There was a marked decline in the catches to 1800 tons after 2012 [20]. A relatively small prawn trawl fishery operates in the Thukela Bank in South Africa, landing a total of 350 tons [21,22]. Prawn trawling in the northern and west coast of Madagascar undertaken by a maximum of 77 vessels in 1996 declined to 53 in 2007, whereas landings varied between 2600 and 4000 tons annually [23]. In Kenya, between 5 and 20 trawlers operated annually within the Malindi–Ungwana Bay, landing between 334 and 640 tons of shallow-water prawns annually during the last few decades [21,23–25]. These fisheries have continued to contribute to coastal economic activities, but with little scientific information on the ecosystem impacts to support their sustainable management.

Conflicts between prawn trawlers and small-scale fishers, as well as environmental concerns in the Malindi–Ungwana Bay (Kenya), resulted in the suspension of the trawl fishery in 2006 by the government pending the development of a management plan to address the social as well as the environmental issues in the fishery [26]. The management plan for the fishery was developed through extensive stakeholder consultations, and regulations for the fishery were enacted in 2010 [27]. The key regulations prescribed in the management plan included restricting the number of vessels to a maximum of four, the mandatory use of turtle excluder devices, regulation on mesh sizes, zoning of fishing area, seasonal bans, restricting trawling time, and submission of a business plan as part of the application for a trawl fishing license. To address environmental concerns, the plan required details for full use of by-catch to be part of the business plan. The plan also introduced a fisheries observer program to collect scientific data and information to evaluate the status of the fishery to support reviews of the regulations in the plan. The fisheries observer program began by deploying observers on four trawlers in 2016 providing an opportunity to evaluate the impacts of the fishery on the ecosystem.

In this study, the catch and effort data (2011–2019) from the fishery and the retained and discarded by-catch data between 2016 and 2019 reported by observers, was evaluated to determine changes in the catch and species composition over time, and the impacts of fishing on the trophic structure of the fish stocks. The results provide information to support the management and planning of fishery to guide sustainable use of the resource.

## 2. Materials and Methods

### 2.1. Study Area

The industrial prawn trawl fisheries operate within the Malindi–Ungwana Bay between latitudes 3°30′ S and 2°30′ S and longitudes 40°00′ N and 41°00′ N, covering the Malindi and Ungwana Bay Complex (Figure 1). The bay is shallow with a wide continental shelf, extending between 15 and 60 km offshore. It is one of the areas suitable for trawling along the Kenyan coast due to the wide continental shelf and absence of coral reefs [28,29]. The benthic habitats are muddy and sandy, some with seagrasses and seaweeds and some rocky areas. The mean depth at high spring tide is 12 m at 1.5 nm and 18.0 m at 6.0 nm. The depth increases rapidly to 100 m after 7 nm and generally decreases northwards. The sub-stratum of the whole of Malindi–Ungwana Bay is mainly composed of siliciclastic sediments [30]. The area has one of the most productive marine fisheries in Kenya as a result of the mangroves forests surrounding the bay, topography of the continental shelf in the bay, and the runoff from the two rivers Sabaki and Tana that drain from a large part of the central and eastern regions of Kenya [31,32] (Figure 1).

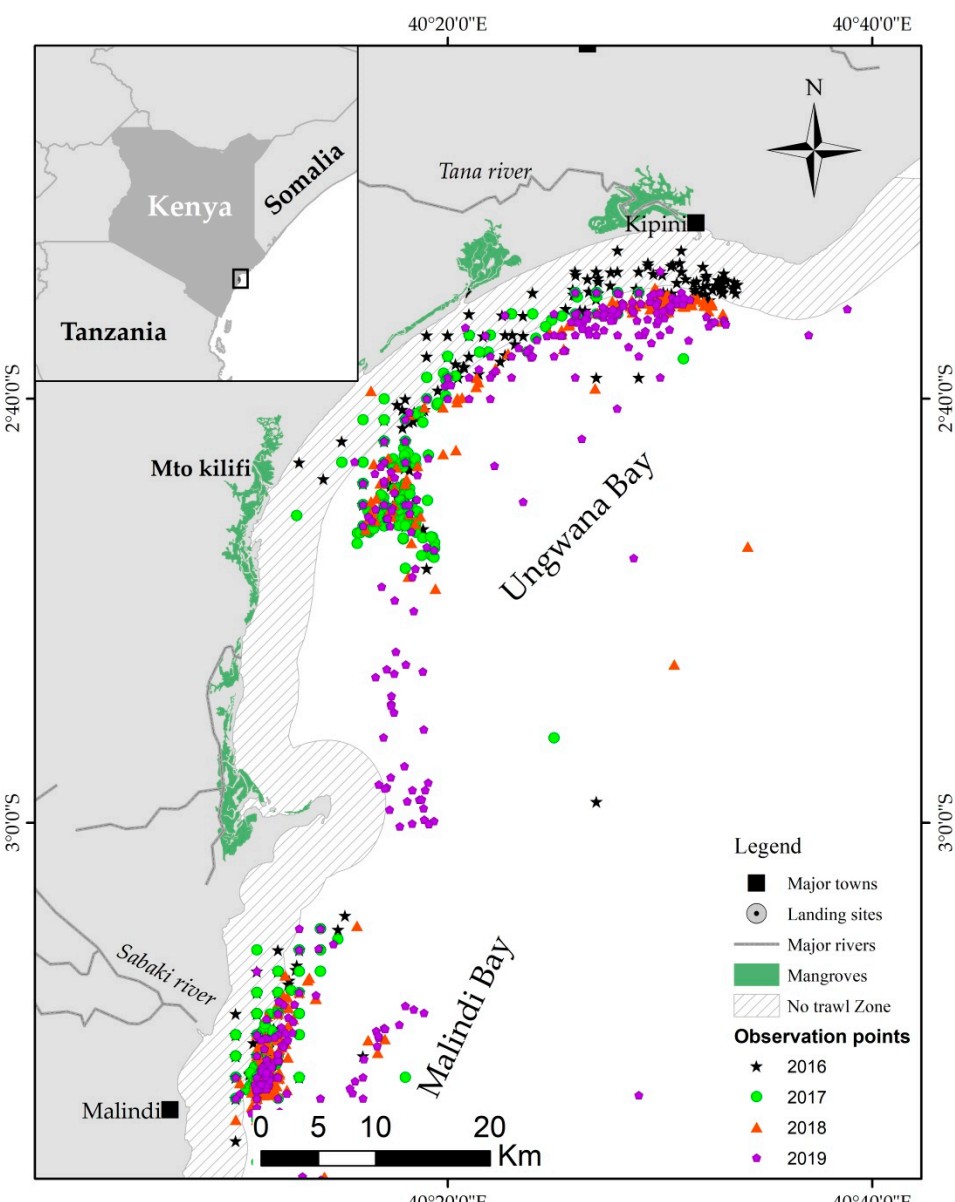

**Figure 1.** Map of Kenya (inset) and the Kenyan coastline showing the location of Malindi and Ungwana Bays, the Sabaki and Tana rivers, and the trawling observations in Malindi–Ungwana Bay from 2016–2019.

The bay is influenced by two dominant offshore current regimes: the Northeast monsoon (NEM) and the Southeast monsoon (SEM). During the SEM, which occurs between April and October, the current circulation is dominated by the northward flow of the East African Coastal Current. During this season, the bay also receives the heaviest river discharge from the Tana and Sabaki Rivers [32]. During the NEM, between November and March, the northward-flowing East African Coastal Current meets the southward-flowing Somali Current to form the Equatorial Counter Current, which flows east into the Indian Ocean [33].

*2.2. Data Collection*

Fisheries and observer data collected on four industrial trawlers licensed to fish in the Malindi–Ungwana Bay during the prawn fishing season between 1 April and 31 October every year from 2016–2019 were used in this study. The trawlers are all Kenyan flagged, with one smaller trawler being 22 m long, 9 m wide, and with a gross registered tonnage

of 117. The three others were 25 m long, 9.3 m wide, and with a gross registered tonnage of 140. All the trawlers had an engine capacity of 300 HP and were fitted with double rigged trawl nets with a mesh size of 55–60 mm and 40–45 mm at the funnel and cod end, respectively. The trawl type was steel beam, with square doors opening 6–15 m, footrope length of 38–200 m, with a net mouth vertical opening of 3–6 m and horizontal opening of 6–20 m, and bottom-line armor of chain. During the shallow-water trawl fishing operations, trawling duration ranged between 2 to 3 h. The captain used GPS and a fish finder installed in the vessel to locate the fishing ground. The observer was positioned on the upper deck of the vessel where he could observe and record the catch and discards following the observer protocol. The catch from each haul was emptied onto a steel sorting table on the deck, large live animals mainly sharks, rays, and turtles were quickly returned to the sea to optimize their chances of survival. These large animals were recorded by the observer. The prawns were collected, graded, cleaned, treated, and packed into 2-kg cartons and blast frozen. The fish were sorted into species to be retained and discarded. The retained fish were put in 25-kg plastic crates, cleaned using pressurized seawater, packed into labeled 5-kg cartons, and blast frozen. As the retained fish were separated from the catch, the remaining unwanted catch was discarded into the sea. The catch that was discarded included small-sized and low-value species. The vessel captains kept a record of the fishing operation and the catch for each haul and sent a weekly report to the Kenya Fisheries Service (KeFS).

Logbook catch data were used to evaluate the catches and fishing effort, between 2011 and 2019. The data consisted of details on each fishing event, including the start and end times and the GPS positions of each haul, catch weights for prawns, finfish, octopus, squids, lobsters, crabs, and others. The catch composition data were collected by scientific observers from the Kenya Marine and Fisheries Research Institute (KMFRI), following sampling protocols adopted from Athayde [34]. At the beginning of each observer trip, the vessel and fishing gear information was recorded. For each fishing event, the start and end fishing positions and times were recorded. Large individuals, including fish, sharks, and rays were first removed from the catch, identified, and recorded. The prawns were sorted from the catch and weighed (kg) on a top-loading balance. The remaining catch was then separated into retained and discards. The retained catch and discards for each haul were weighed and sub-samples for identification were collected. The individuals in the samples were separated into species following standard species identification guides for the region [35–38], counted, and weighed. The catch composition was obtained by multiplying the sample data with the raising factor (i.e., number of portions of which main catch was divided), using the catch composition estimation method. The total catch weight was obtained, by adding the total weight of non-random samples (large fish put aside) and the scaled-up weights obtained from the samples. The data were recorded in a standard data sheet developed for the observer program.

On disembarking from a completed observer trip, an observer coordinator verified the data during a debriefing session with the observer to correct any mistakes before the data were entered into a spreadsheet. The data were cleaned by making sure that all names of species were correct, and the dates, times, GPS positions, and weights were correctly entered in standard units.

*2.3. Coverage of Fishery Observers*

Thirty-seven observers were deployed between 2016 and 2019 and recorded a total of 1371 hauls. Observations were undertaken in all the months of the shallow-water prawn fishing season in 2017, while a few months of the fishing season were not observed in 2016, 2018, and 2019 (Table 1). Overall, between 11% and 19% of the fishery was observed every year. The trawling observations taken from 2016 to 2019 are shown in Figure 1.

**Table 1.** Number of observers, deployments, and hauls during the study period.

| Year | Months Observed | Number of Observers | Number of Deployments | Total Number of Trawls | Units of Trawls Recorded | % Observed Trawls |
|------|-----------------|---------------------|-----------------------|------------------------|--------------------------|-------------------|
| 2016 | Jun, July, Aug, Sept, Oct | 7 | 9 | 1843 | 318 | 17.3 |
| 2017 | Apr, May, Jun, Jul, Aug, Sep, Oct | 6 | 10 | 1963 | 376 | 19.2 |
| 2018 | May, Jun, Jul, Aug, Oct | 6 | 7 | 2400 | 281 | 11.7 |
| 2019 | Aug, May, Jun, Jul, Aug, Sep, Oct | 9 | 11 | 2325 | 396 | 17.0 |

### 2.4. Data Analysis

The variation in the nominal total catch, discarded, and retained catch was evaluated using time-series graphs. Nominal catches were used to allow comparisons with previous studies, which also used nominal catches. The species composition of the retained and discarded catches was described using two metrics: species diversity (Shannon) and mean trophic level. The trophic level for each species was obtained from FISHBASE [39]. The mean trophic level was calculated as:

$$TL_L = \sum_{i=1}^{n} Y_i \cdot TL_i / Y_L \tag{1}$$

where $Y_i$ is the catch of species $i$, TL is the trophic level of species $i$.

The Mann–Whitney U-test was then used to compare the differences in species diversity and the mean trophic level of the total catch in the two bays for the four years (2016–2019).

The species composition of the catches was investigated using nonmetric multidimensional scaling (nMDS) ordination on standardized and square root transformed data to compare differences between the months, years, and retained vs. discarded species. A hierarchical group-average clustering based on a Bray–Curtis similarity matrix was overlaid to elucidate similarities between seasons, depth, and sites [40]. The relative distance of the data points provided a measure of similarity. A posterior analysis of similarity (ANOSIM) test was applied to check for significant differences in the species composition between years, retained and discarded species, followed by a similarity of percentage (SIMPER) analysis, which identified species that contributed most to dissimilarities between the years, retained and discarded species. The statistical analyses were conducted using STATISTICA (StatSoft, Inc., Tulsa, OK, USA) and PRIMER [41].

### 3. Results

#### 3.1. Catch and Effort

The annual trends in fishing effort and total catches obtained from the fishery data between 2011 and 2019 showed increasing catches with an increase in fishing effort over time (Figure 2). The fishing effort increased tenfold from 437 h in 2011 to 5102 h in 2019, with the steepest increase from 2013 to 2016. The by-catch increased more than 20 times within the nine years, from 20 tons in 2011 to 450 tons in 2019. Prawn catches also increased more than 20 times with increasing effort, but the increase was gradual, from 6 tons in 2011 to 133 tons in 2019.

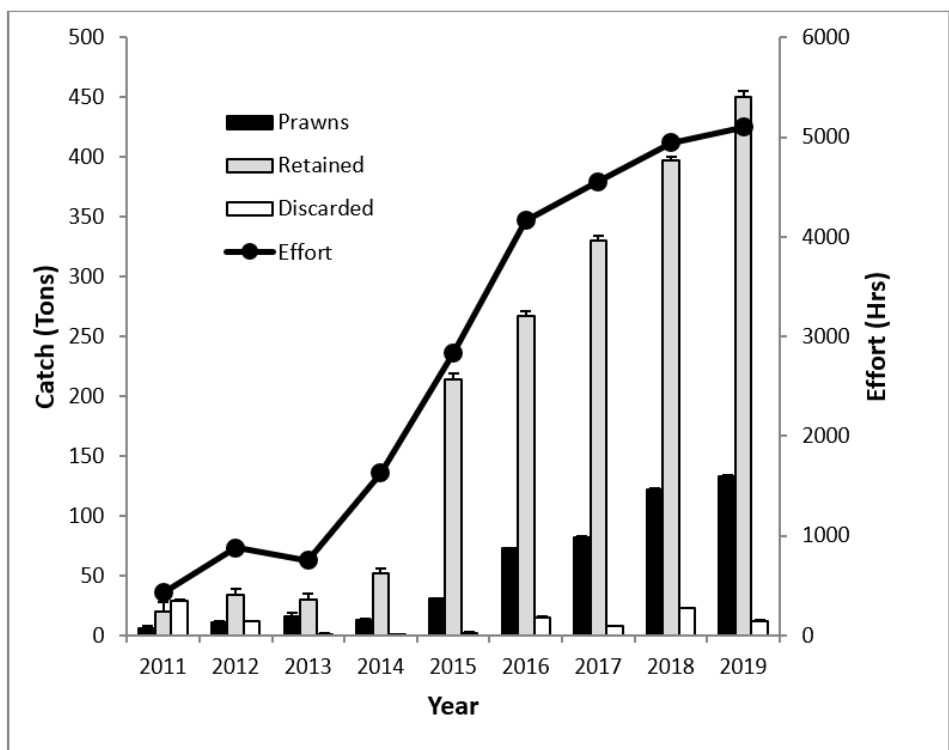

**Figure 2.** Annual trends in trawl catch and effort (±SE) in Malindi–Ungwana Bay in the 2011–2019 period.

### 3.2. Spatial and Seasonal Variation in the Catch

The Bray–Curtis similarity analysis of species composition between the seasons, depth, and site (Figure 3) showed a significant difference in the depth, with the depth of >60 m being different from the other depths (0–20, 21–40, and 41–60). There were no significant differences in species composition between seasons and sites.

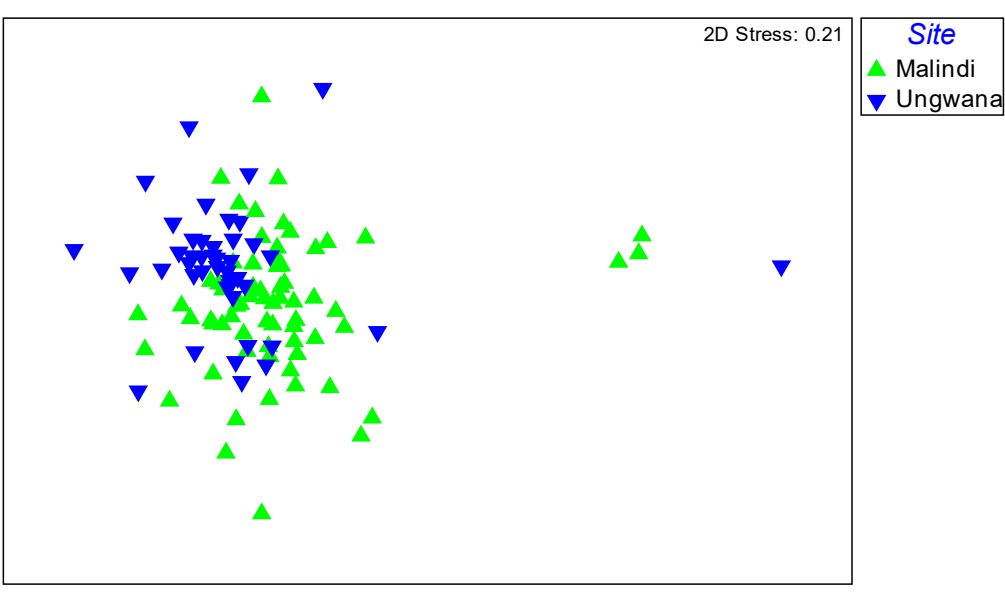

(**a**)

**Figure 3.** *Cont.*

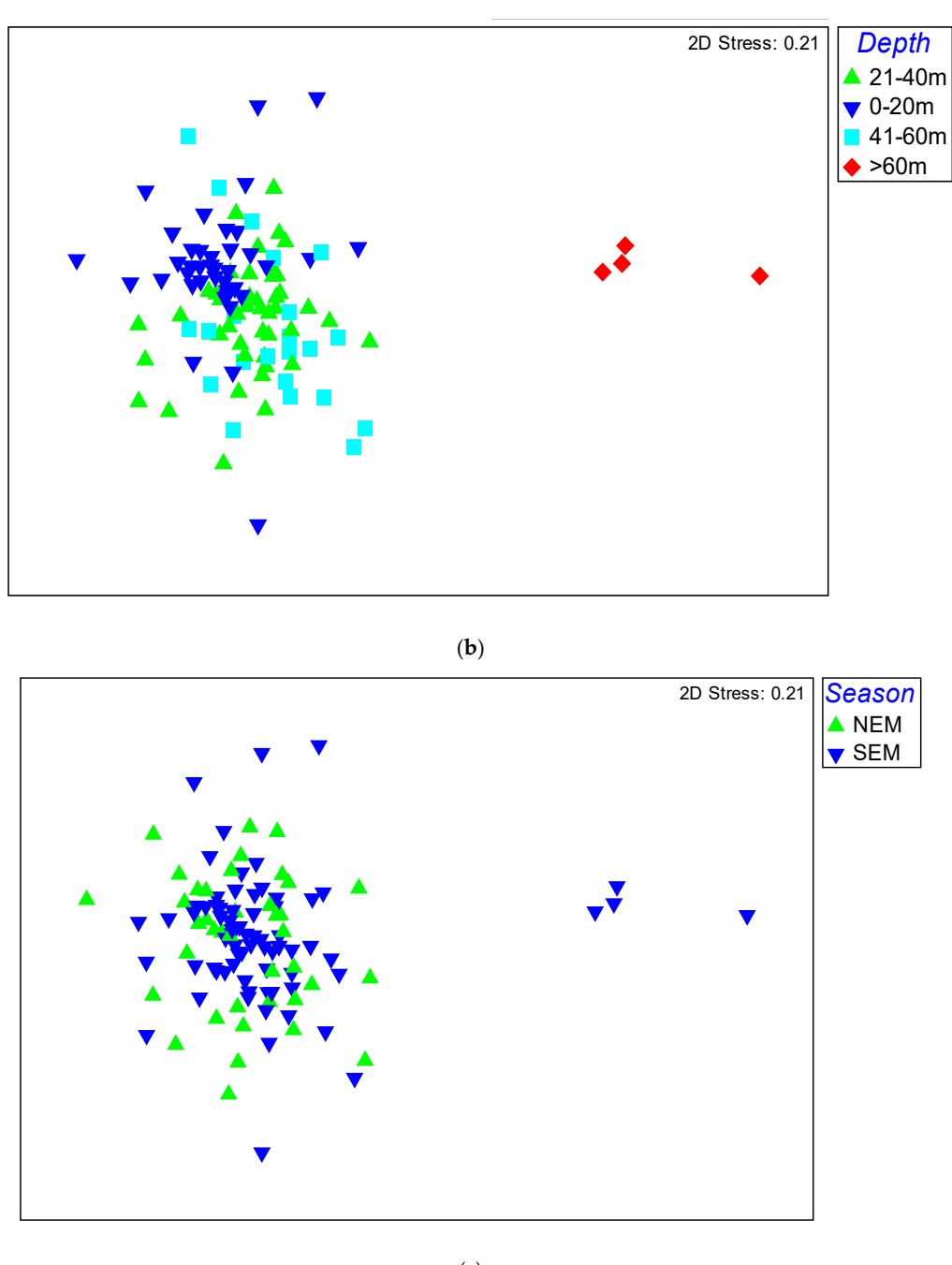

(**b**)

(**c**)

**Figure 3.** Bray–Curtis similarity plots of species composition between the (**a**) sites, (**b**) depth, and (**c**) seasons.

### 3.3. Variation in Diversity and Trophic Levels

There were no significant differences in the species diversity between the two bays and the years (Mann–Whitney U-test, U = 8, P = 1.00). The annual Shannon diversity index for both bays together ranged from 2.6 (±0.10 SE) in 2018 to 3.0 (±0.1 SE) in 2017. Malindi Bay had the highest species diversity of 3.1 (±0.15 SE) in 2017 and the lowest species diversity of 2.6 (±0.12 SE) in 2018 (Figure 4). Ungwana Bay had the highest species diversity (2.9 ± 0.13 SE) in 2017 and 2019.

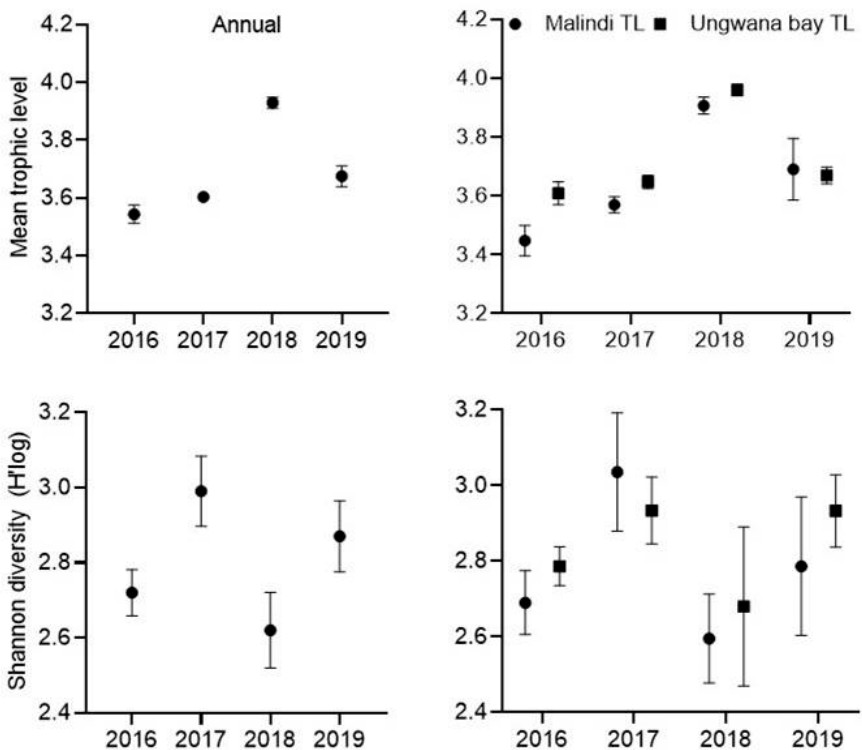

**Figure 4.** Annual mean trophic level and mean species diversity: general and on each bay. Error bars: SE.

The results of the Mann–Whitney U-test on the mean trophic levels showed no significant differences between the two bays and the years (Mann–Whitney U-test, U = 6, P = 0.739). The annual mean trophic level for both bays combined from 2016 to 2019 ranged from 3.5 ($\pm$0.03 SE) to 3.9 ($\pm$0.02 SE). The highest mean trophic level was in 2018. The mean trophic levels of the two bays were similar from 2016 to 2019 with Ungwana Bay having slightly higher mean trophic levels than Malindi Bay. Both bays had a high mean trophic level in 2018 (Figure 4).

### 3.4. Retained and Discarded Catch

The relative amounts of catch recorded by observers showed that retained catch increased between 2016 and 2019 (Figure 5). The discarded catch was highest in 2017 but gradually decreased in 2018 and 2019. The overall target prawn: total by-catch ratio was 1:9 from 2017 to 2019 compared to 1:3 during 2016. The prawn: discarded catch ranged from 1:1.7 (2016) to 1:3.3 (2017). The trends in the target catch, retained by-catch, and discarded by-catch indicated a relative reduction in target catch (Figure 5). On average the total catch comprised 16% target, 59% retained, and 25% discards. The proportion of the target prawns was highest in 2016 but declined through the other years (Figure 5).

### 3.5. Composition of Retained and Discarded Catch

Overall, 475 species were recorded by observers during the study period with the highest number of species (275) recorded in 2019. Among the top 10 retained and discarded species in terms of weight, *Otolithes ruber* and *Pomadasys maculatus* were captured in all years, with *O. ruber* comprising 10 to 20% of the retained catch (Table 2). None of the targeted prawn species were in the list of top-ten retained species. Three species *Pellona ditchela*, *Galeichthys feliceps,* and *Secutor insidiator* were captured in the top 10 of the discarded species in all years, with *P. ditchela* being the most discarded species (comprising 14% of discards). Some species retained also appeared in the list of discards, e.g., *P. ditchela*, *S. insidiator*, *G. feliceps*, *Trichiurus lepturus,* and *Leiognathus equulus*.

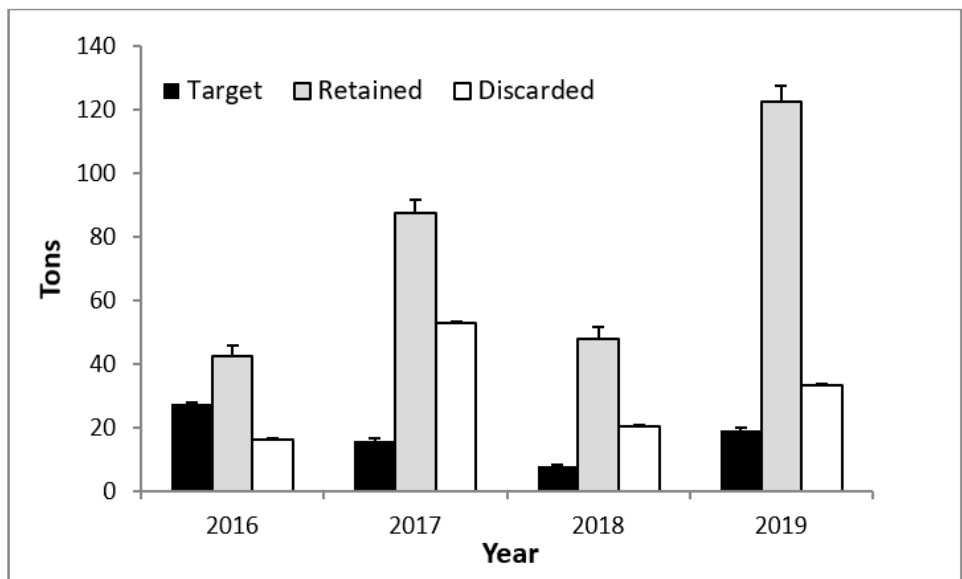

**Figure 5.** Trawl-fishery target retained and discarded catch (±SE) recorded by observers between 2016 and 2019 in Malindi–Ungwana Bay.

**Table 2.** Top-ten species by weight of retained and discarded species recorded from 2016 to 2019. Weight in kg (percentage).

|  | 2016 | 2017 | 2018 | 2019 |
|---|---|---|---|---|
| Retained |  |  |  |  |
| *Otolithes ruber* | 8581.48 (20.37) | 9183.81 (10.5) | 6505.43 (13.35) | 13,649.81 (11.12) |
| *Pomadasys maculatus* | 3063.06 (7.27) | 3687.24 (4.22) | 2711.51 (5.56) | 5777.21 (4.71) |
| *Galeichthys feliceps* | 2770.25 (6.58) | 2760.77 (3.16) |  |  |
| *Johnius dussumieri* | 2579.50 (6.12) |  |  |  |
| *Lobotes surinamensis* | 2167.08 (5.14) |  |  |  |
| *Upeneus sulphureus* | 1965.95 (4.67) | 5326.79 (6.09) |  |  |
| *Sphyraena jello* | 1695.13 (4.02) | 2667.26 (3.05) | 1835.36 (3.82) |  |
| *Leiognathus equulus* | 1531.83 (3.64 | 3360.49 (3.84) |  |  |
| *Trichiurus lepturus* | 1381.24 (3.28) |  |  |  |
| *Pomadasys multimaculatus* | 1335.00 (3.17) |  |  |  |
| Discards |  |  |  |  |
| *Pellona ditchela* | 2393.18 (14.6) | 7605.41 (14.33) | 2695.23 (13.1) | 4133.17 (12.35) |
| *Trichiurus lepturus* | 1729.85 (10.55) | 4465.77 (8.41) | 1119.30 (5.44) |  |
| *Galeichthys feliceps* | 1531.19 (9.34) | 2634.68 (4.96) | 1198.54 (5.83) | 1570.19 (4.69) |
| *Secutor insidiator* | 1394.63 (8.51) | 5668.83 (10.68) | 2206.81 (10.73) | 2226.35 (6.65) |
| *Photopectoralis bindus* | 1177.52 (7.18) | 1418.39 (2.67) |  |  |
| *Nematopalaemon tenuipes* | 1078.18 (6.58) |  | 1347.20 (6.55) |  |
| *Thryssa vitrirostris* | 982.82 (6) |  | 1257.57 (6.11) | 1136.33 (3.4) |
| *Thryssa malabarica* | 567.22 (3.46) |  |  |  |
| *Leiognathus equulus* | 407.09 (2.48) |  | 2543.74 (12.37) | 1850.31 (5.53) |
| *Johnius amblycephalus* | 371.22 (2.27) | 1527.96 (2.88) |  |  |
| *Secutor ruconius* | 2554.29 (4.81) |  |  |  |

The result of the nMDS ordination analysis of the catch data recorded by observers showed a clear difference between the compositions of retained and discarded species, an indication that the selection is not random (Figure 6). ANOSIM revealed a strong dissimilarity between the composition of retained and discarded species (R = 0.709, *p* = 0.001). Three species were most responsible for 86.38 of the average dissimilarity between retained and discarded species: *P. ditchela*, *Nematopalaemon tenuipes,* and *S. insidiator* (Table 3).

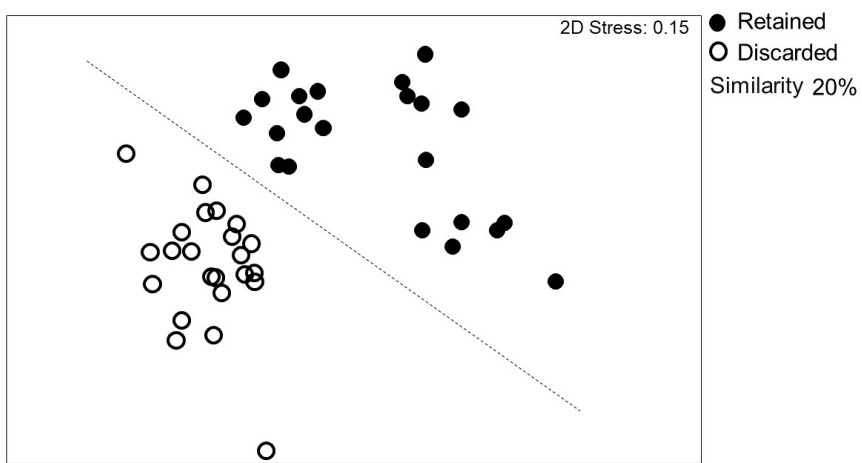

**Figure 6.** Non-metric multidimensional scaling ordination plots of retained vs. discarded species.

**Table 3.** Summary of SIMPER analysis showing the average dissimilarity in the species composition between retained and discarded catch and three species that contributed most to the overall dissimilarities.

| Species | Average Abundance (%) | | Dissimilarity | Contrib% | Cum.% |
|---|---|---|---|---|---|
| | Retained | Discarded | Av.Diss = 86.38 | | |
| *Pellona ditchela* | 1.11 | 3.5 | 2.83 | 3.28 | 3.28 |
| *Nematopalaemon tenuipes* | 0.01 | 2.3 | 2.58 | 2.99 | 6.27 |
| *Secutor insidiator* | 0.39 | 2.66 | 2.49 | 2.88 | 9.15 |

The nMDS ordination of the catches between years, grouped 2016 and 2017 as more similar in species composition (ANOSIM, R = 0.23) influenced by the composition of retained species (Figure 7), while all derived pairwise comparisons with 2018 and 2019 catches were strongly dissimilar (ANOSIM R values of 0.99). However, the observed dissimilarities between years were not statistically significant ($p > 0.05$). The species composition of discarded species did not have a clear pattern as that of the retained catch. The discarded species in 2016 and July and October 2018 were dissimilar from those of 2017, some months of 2018, and 2019 (Figure 8). Discarded species in September 2017 and May 2019 were similar, while those in August 2017 and July 2019 were dissimilar to all the other observed months (Figure 8).

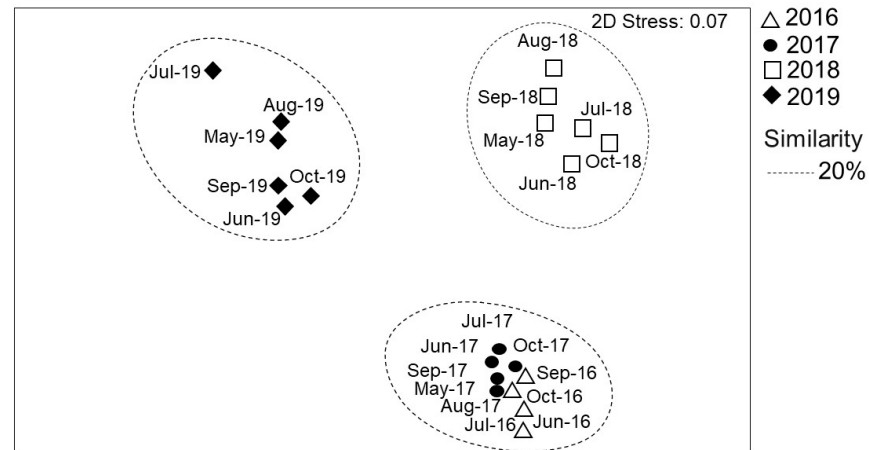

**Figure 7.** Non-metric multidimensional scaling ordination plots for retained catches in the months of different years.

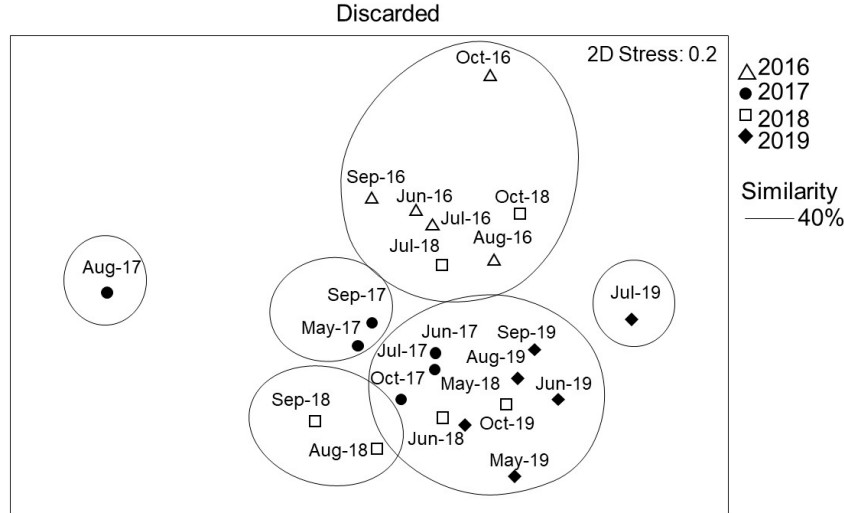

**Figure 8.** Non-metric multidimensional scaling ordination plots for discarded catches in the months of different years.

Result of SIMPER for the retained species identified *O. ruber*, *Panulirus homarus*, *Penaeus indicus*, and *Penaeus japonicus* as most responsible for the dissimilarity between 2016 and the other years (2017, 2018, and 2019), while *P. homarus* was most responsible for the dissimilarity between 2019 and the other years (2016, 2017, and 2018), contributing between 4.4 and 4.8% (Table 4). For discarded species, *N. tenuipes* contributed most to the dissimilarity between 2016 and the other years, contributing between 4 to 11% (Table 5).

**Table 4.** Summary of SIMPER analysis showing the average dissimilarity in the composition of retained species across the years studied and three species that contributed most to the overall dissimilarities.

| Species | Average Abundance (%) | | Dissimilarity | Contrib% | Cum.% |
|---|---|---|---|---|---|
| | 2016 | 2017 | Av.Diss = 50.63 | | |
| *Upeneus vittatus* | 0.6 | 1.68 | 1.51 | 2.98 | 2.98 |
| *Pellona ditchela* | 1.89 | 1.92 | 1.42 | 2.81 | 5.79 |
| *Plesionika martia* | 1.26 | 0.37 | 1.36 | 2.69 | 8.48 |
| | 2016 | 2018 | Av.Diss = 81.98 | | |
| *Otolithes ruber* | 3.62 | 0.06 | 3.49 | 4.25 | 4.25 |
| *Penaeus japonicus* | 0.68 | 3.9 | 3.2 | 3.91 | 8.16 |
| *Galeichthys feliceps* | 2.72 | 0.02 | 2.64 | 3.23 | 11.38 |
| | 2016 | 2019 | Av.Diss = 86.74 | | |
| *Panulirus homarus* | 0 | 3.89 | 3.78 | 4.36 | 4.36 |
| *Penaeus indicus* | 3.64 | 0 | 3.43 | 3.95 | 8.31 |
| *Otolithes ruber* | 3.62 | 0 | 3.41 | 3.93 | 12.24 |
| | 2017 | 2018 | Av.Diss = 79.88 | | |
| *Penaeus japonicus* | 0.87 | 3.9 | 2.95 | 3.69 | 3.69 |
| *Metapenaeus stebbingi* | 0.26 | 3 | 2.62 | 3.28 | 6.97 |
| *Penaeus indicus* | 3.64 | 1.14 | 2.54 | 3.18 | 10.15 |
| | 2017 | 2019 | Av.Diss = 83.23 | | |
| *Panulirus homarus* | 0 | 3.89 | 3.73 | 4.49 | 4.49 |
| *Penaeus indicus* | 3.64 | 0 | 3.42 | 4.11 | 8.6 |
| *Uroconger lepturus* | 0 | 2.51 | 2.44 | 2.93 | 11.53 |
| | 2018 | 2019 | Av.Diss = 82.48 | | |
| *Panulirus homarus* | 0 | 3.89 | 3.96 | 4.8 | 4.8 |
| *Penaeus japonicus* | 3.9 | 1 | 2.99 | 3.63 | 8.43 |
| *Metapenaeus stebbingi* | 3 | 0.23 | 2.72 | 3.3 | 11.72 |

**Table 5.** Summary results of SIMPER showing the average dissimilarity in the discarded species composition across the years studied and three species that contributed most to the overall dissimilarities.

| Species | Average Abundance (%) | | Dissimilarity | Contrib% | Cum.% |
|---|---|---|---|---|---|
| | 2016 | 2017 | Av.Diss = 68.29 | | |
| *Nematopalaemon tenuipes* | 6.81 | 0.73 | 7.48 | 10.95 | 10.95 |
| *Thryssa setirostris* | 0.67 | 2.7 | 2.55 | 3.73 | 14.69 |
| *Trichiurus lepturus* | 2.2 | 3.03 | 2.52 | 3.69 | 18.38 |
| | 2016 | 2018 | Av.Diss = 64.54 | Contrib% | Cum.% |
| *Nematopalaemon tenuipes* | 6.81 | 2.25 | 6.87 | 10.65 | 10.65 |
| *Leiognathus equulus* | 0.82 | 1.98 | 2.35 | 3.65 | 14.3 |
| *Secutor insidiator* | 1.74 | 3.24 | 2.34 | 3.63 | 17.93 |
| | 2016 | 2019 | Av.Diss = 69.51 | Contrib% | Cum.% |
| *Nematopalaemon tenuipes* | 6.81 | 0.17 | 8.09 | 11.63 | 11.63 |
| *Pellona ditchela* | 2.6 | 4.29 | 2.44 | 3.5 | 15.14 |
| *Thryssa setirostris* | 0.67 | 2.38 | 2.17 | 3.12 | 18.25 |
| | 2017 | 2018 | Av.Diss = 65.07 | Contrib% | Cum.% |
| *Nematopalaemon tenuipes* | 0.73 | 2.25 | 3.07 | 4.72 | 4.72 |
| *Trichiurus lepturus* | 3.03 | 2.08 | 2.36 | 3.63 | 8.34 |
| *Antennarius pictus* | 2.05 | 0 | 2.34 | 3.59 | 11.93 |
| | 2017 | 2019 | Av.Diss = 66.28 | Contrib% | Cum.% |
| *Trichiurus lepturus* | 3.03 | 1.34 | 2.4 | 3.62 | 3.62 |
| *Pellona ditchela* | 2.94 | 4.29 | 2.34 | 3.53 | 7.15 |
| *Antennarius pictus* | 2.05 | 0 | 2.14 | 3.22 | 10.38 |
| | 2018 | 2019 | Av.Diss = 61.76 | Contrib% | Cum.% |
| *Nematopalaemon tenuipes* | 2.25 | 0.17 | 2.86 | 4.63 | 4.63 |
| *Gazza minuta* | 0.16 | 1.94 | 2.2 | 3.56 | 8.19 |
| *Leiognathus equulus* | 1.98 | 2.28 | 2.19 | 3.54 | 11.73 |

## 4. Discussion

### 4.1. Catch and Effort

This study analyzed the catch trends using fisheries logbook data and observer data with the aim of determining the impact of fishing on the ecosystem. The catch and effort increased during the nine years of operation with the prawn and fish catch ranging from 6 to 133 tons and 20 to 450, respectively, from 2011 to 2019. Previously reported average annual landings in the bay were higher between 257 and 400, and 315 and 602 tons of prawns and fish, respectively, between 2001 and 2006 [42]. The average prawn landings in the Malindi–Ungwana Bay were 400 tons per year in the 1970s, 1980s, and 1990s [43]. Thus, the prawn catch is yet to reach these earlier reported catches. The lower total annual catches could be attributed to the restricted fishing effort, with six months annual fishing season and only during the daytime from 6 a.m. to 6 p.m. required by the management plan regulations. In addition, the regulations in the management plan zoned < 3 nm offshore as no trawling to reduce the interaction of trawlers with small-scale fishing gears. Stock surveys have shown that prawn stocks are higher close to the shore and the estuaries [42,44]. The zoning in the management regulations moved the trawlers from the centers of prawn stocks concentration resulting in the lower catch.

### 4.2. Spatial and Seasonal Variation

Overall, in this study, no significant differences in the species composition between the seasons and the two bays (Malindi and Ungwana) were found. However, distinct abundance and species composition patterns have been reported in the same bay for prawns driven by the bottom type and depth [45]. However, depth showed a significant difference in the species composition at depths > 60 m. These findings indicate that there were no changes in the species composition over the four years of observations, and the

dominant species in the catches have remained the same. However, the species composition in the bay is influenced by the depth.

### 4.3. Diversity and Trophic Levels

Species diversity in the catch did not vary significantly and was dominated by a few species and families, similar to other trawl fisheries in the tropics [46,47]. A previous study reported *G. feliceps*, *O. ruber*, *Johnius amblycephalus*, *Johnius dussumieri*, *Lobotes surinamensis*, *L. equulus*, *P. maculatus* as the dominant species in the trawl catches of Malindi–Ungwana Bay [48]. These species were also reported as dominant in this study.

Results of the nMDS analysis indicated that species composition from prawn trawling differed over the four years. The species contributing to differences in the retained catch in the four years were *P. indicus*, *O. ruber*, and *P. homarus* with *Otolithes ruber* being most abundant. In particular, *P. homarus* contribute to the dissimilarity of the catches in 2019 indicating a shift in the distribution of fishing effort to deeper water. Based on numbers, *N. tenuipes* and *P. ditchella* were responsible for the difference in species composition of the discarded catch. *Nematopalaemon tenuipes* is discarded because it is considered to be too small and of low economic value [49]. *Pellona ditchela* also appeared in the top 10 species in terms of weight. In a previous study, *G. feliceps* and *P. ditchela* were the most dominant species contributing to the highest spatial dissimilarity in the inshore areas of the bay [50]. Overall, in this study, no significant differences in the species composition between the seasons and the two bays (Malindi and Ungwana) were found. However, depth showed a significant difference in the species composition at depths > 60 m. These findings indicate that there were no changes in the species composition over the four years of observations, and the dominant species in the catches have remained the same. However, the species composition in the bay is influenced by the depth.

Mean trophic levels indicate the status of resource exploitation and is an indicator of fishery-induced impacts at the food web level [51,52]. The trophic levels in this study (3.45–3.96) were comparable with the values between 3.2–4 recorded in an earlier study [48], an indication of no marked variation and a relatively stable ecosystem, meaning that the impacts of fishing at the current level has not surpassed the self-regulatory capacity of the bay [51,53].

### 4.4. Retained and Discarded Catch

Globally, the trawl fishery is characterized by a high by-catch rate, with prawn trawling reporting prawn to by-catch ratios of between 1:3 to 1:15 [54,55]. In this study, the prawn to by-catch ratio decreased through the four years and the prawn to discarded catch ratio of 1:1.7 to 1:3.3 is comparable to the prawn to discarded by-catch ratio of 1:1.5 obtained in 2012 [42]. The by-catch in this study made about 84% of the catch of which 59% comprised retained by-catch and 25% of discarded by-catch. Other studies estimated the total by-catch from prawn trawling in the Malindi–Ungwana Bay to be 70–80% by weight [43]. In Mozambique, the by-catch comprised about 80% of the total catches [21]. These estimates of by-catch are comparable with those found in this study and are an indication of high amounts of by-catch resulting from prawn trawling.

The reasons for discarding by-catch are attributed to lack of storage space and the low value of small fish [56]. The discarded by-catch reported for the Malindi–Ungwana Bay comprised different families, mainly the Leiognathidae, Clupeidae, Dasyatidae, and Carcharhinidae (This study, [43]). Non-commercial fishes contributed more than 43% of the discards, whereas juveniles of some commercially important species, such as *O. ruber* and *Johnius* sp. (Sciaenidae), and *Pomadasys* sp. (Haemulidae), made up 25% of discards [43,57]. In comparison to our study, the families that were discarded were mainly Pristigasteridae, Trichiuridae, and Ariidae while the species retained included *O. ruber*, *P. maculatus*, *G. feliceps* among others. Thus, the commercially important species that were previously being discarded were now being retained. Species that were previously discarded may have

gained acceptance and value in the local market making retaining them cost-effective. The fishing industry has been exploring ways of maximizing the use of by-catch.

### 4.5. Composition of Retained and Discarded Catch

Overall, the number of all species reported in this study increased with fishing effort over the years, with the highest 275 in 2019. The authors of [48] reported 223 species, while in a bottom trawl survey in the bay in 2012, 66 fish species in 43 families were found with the highest biomass in the shallow areas (<50 m) [58]. In an earlier survey, the number of species collected was 160 species belonging to 61 families [44]. This shows that the number of species reported varied from the different studies, with the highest reported in this study attributed to the length of time over which the fishery observations were made.

Fish species contributed higher biomass compared to the target prawns, which were ranked below the 10 top species. This is the general observation in most prawn trawl fisheries of the world where large amounts of non-target species are caught [59,60]. In Malindi–Ungwana Bay, of the five target penaeid prawns, *P. indicus* is the most dominant [61]. Though seasonal variations influence prawn catches, *P indicus* along with *P. monodon*, *P. monocerous*, *P. semisulcatus*, and *P. japonicus* are common prawn species in the bay [25,49,50,62].

Concerns have been raised regarding the overfishing by trawlers of species important in small-scale fisheries in the Western Indian Ocean (WIO) region, including *O. ruber* [63]. The species is a common trawl fishery by-catch along the East African coast and is usually retained for its high commercial value. Reduced abundance of *O. ruber* along with other common species associated with prawn trawling could result in ecological changes (e.g., altered predator–prey relationships) and impact the artisanal fishers' catches [63]. Predator–prey relationships between finfish and prawns may contribute to the resulting high abundance of finfish by-catch [64]. Most of the finfish abundant in the by-catch, such as *O. ruber*, *P. maculatus*, *P. ditchela*, *Thryssa vitrirostris*, and *L. equulus*, *Terapon jarbua*, are predators of penaeid prawns [64,65].

### 4.6. Reduction and Use of By-Catch

Discarded catch from prawn trawling has been a concern for a long time, with pressure to reduce the capture of non-commercial species increasing [5,60]. The by-catch in prawn trawling can be reduced but cannot be eliminated, and it is estimated that present selectivity technology and management can reduce by-catch by 30 percent at most [21]. By-catch reducing devices are increasingly being used in prawn trawling to reduce the amounts of by-catch, and in some areas, they have been successful and beneficial to prawn fisheries [60,66–68].

Besides the efforts to reduce the by-catch, the complete utilization of catch is also considered an important way of increasing the benefits from the fisheries [69]. In China, the by-catch was used for the aquaculture industry [70]. In Madagascar, by-catch is normally sold for human consumption [69]. In Kenya, the retained by-catch is offloaded in urban centers, mainly Malindi and Mombasa, and sold to women in the fish retail business (locally known as "*Mama karanga*") [71]. Increasing the amount of by-catch that reaches the market would support more livelihoods through trade and support food security, particularly in the urban centers within the coast region. The utilization of by-catch as food reduces the ethical argument against the un-selective fishing of trawl fisheries.

### 5. Conclusions

The Malindi–Ungwana Bay fishery is a good example of the competing interests of fisheries resource use and ecosystem conservation needs between resource users; including industrial fishing, small-scale fishing, as well as recreational use of the ecosystem, in which industrial fishing has been criticized for environmental degradation and large wastage in form of discards. Consequently, several studies have been undertaken to assess the Malindi–Ungwana Bay fishery addressing the status of the fishery [49,61], ecological indicators affecting the fishery [72], and resource use and distribution [42,50]. This study

evaluated the state of the shallow-water prawn trawl fishery of the Malindi–Ungwana Bay based on some catch and ecosystem-based indices, after the implementation of the management plan in 2010. The results showed an increase in fishing effort and catch over the four years, 2016 to 2019. The levels of by-catch remained high, while the proportion of retained by-catch increased over the years. The species composition of the trawl catches in the two bays was similar and the dominating species in the Malindi–Ungwana Bay remained the same over the years. There was a distinct difference between the retained and discarded species, and differences in species composition of retained catch over time. However, the evaluation showed no marked decline in the status of the stock in the bay based on the diversity and tropic indices. We recommend that more of the discarded species are retained to ensure that the fishery is less wasteful.

**Author Contributions:** Conceptualization, E.N.F. and E.N.K.; Data curation, E.N.F., J.O.O., N.W., P.T. and E.N.K.; Formal analysis, E.N.F., J.O.O., G.M.O. and E.N.K.; Methodology, E.N.F., J.O.O., G.M.O., P.T. and G.W.M.; Validation, E.N.F., J.O.O., N.W., E.N.K., G.M.O. and G.W.M.; Writing—original draft, E.N.F.; Writing—review and editing, E.N.F., J.O.O., N.W., G.M.O., P.T., G.W.M. and E.N.K. All authors have read and agreed to the published version of the manuscript.

**Funding:** This research received no external funding.

**Institutional Review Board Statement:** Not applicable.

**Data Availability Statement:** The data presented in this study are available on request from the corresponding authors. The data are not publicly available due to confidential information included in the data.

**Acknowledgments:** We are grateful to the Kenya Marine and Fisheries Research Institute (KMFRI), Kenya Fisheries Service (KeFS), and the fishing industry (East African Sea Foods Ltd. and Ittica Ltd., Mombasa Kenya) that provided the logistical support of all the shallow-water prawn trawling observer program surveys. We appreciate the cooperation and support from the captains of the fishing vessels and the entire fishing crew who provided the scientific observers with the necessary support to collect data and samples. The support, cooperation, and commitment of KMFRI fishery observers (Boaz Orembo, Masudi Juma Zamu, Rashid Anam, James Gonda, Jibril Olunga, Ben Ogola, Sammy Kadhengi, Benard Kimathi, Geoffrey Odhiambo Otieno, Justus Andati, Nimrod Ishmael, Hafidh Ishmael) during the field and laboratory work are appreciated. We are grateful to our respective institutions (KMFRI and The Nature Conservancy) for supporting authors' time in this study. We also acknowledge the reviewers who gave constructive changes and recommendations to improve the manuscript.

**Conflicts of Interest:** The authors declare no conflict of interest.

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
