# Peer review of "Diversity of Shallow-Water Species in Prawn Trawling: A Case Study of Malindi–Ungwana Bay, Kenya"

_diversity, doi:10.3390/d14030199_

Round 1

Reviewer 1 Report

This is an important contribution to several issues, including the impacts of shrimp trawling and its tensions with conservation and artisanal fishing. These issues have clearly high local relevance, but the results presented here are also applicable to a wider regional and even global scale. To maximize its relevance, the paper needs to be further developed in the areas below.

1- the authors must clearly separate the analysis of (i) the effects of trawling from that of (ii) the uses given to the bycatch. This should be reflected in the organization of the paper. The first topic should be studied exploring the temporal variation of fishing effort and the resulting catch, divided into target and bycatch. Included here are, obviously, the issues of species diversity and trophic level, but all calculated based on the whole catch or its taxonomic subdivisions. Specifically, this section should never use the discarded/retained subdivision. This subdivision should be approached on a specific section of the paper, dedicated to the issue of what species and what percentage of bycatch are retained, given the desirable objective of eliminating the discards. 

2- more attention should be given to the readability of the paper. For this, authors should make sure that each paragraph (i) covers only one idea and (ii) has the standard structure:  a first sentence giving the main point to be made, followed by supporting sentences providing evidence or explanations, and an optional concluding sentence, connecting the point to the overall argument.

3- make sure that all results are presented in the respective section so that the discussion can focus on the analysis. Examples to be corrected include the reference to inconsistencies between the two datasets (lines 297-300) and the species composition of the prawn catch (lines 351-352).

Further comments and suggestions were annotated in the manuscript

Reviewer 2 Report

The manuscript is well designed amd contain original information of the problem that is interesting for marine ecologists. I suggest to accept the manuscript for publication. 

Author Response

Thank you for reviewing our manuscript.

Reviewer 3 Report

This is valuable paper based on the large amount of fisheries data, from the data-deficit region.

It shows, how in the the course of 8 years, and introduction the fishery  managament, the increase fishing effort and catches with little effect on the diversity of chatch was observed.

I guess that paper needs only minor corrections – from the environmental- conservation point of view.

  • The title is a bit misleading: „ Impact of trawling on the diversity of shallow water communites: A case study of Malindi-Ungwana bay, Kenya” while it presents only the diversity of catches – species collected during the fishing activity. So, may be „”…diversity of shallow water species in fish gear” ?
  • The description of the methods – fishing gear lack the size of the trawl (width, horizontal opening in metres) and type of the bottom line armour – steel bobines? Chain ? soft rope ?
  • Some words are needed about benthic habitats – is it flat sandy bottom without structural species (seeweeds, Hydrozoa, Bryozoa, corals, long living polychaetes etc..) or some structural forms exists ?
  • The discarded part of the catch – was it because of the damaged specimens (as some valuable species were discarded) or it was impossible to process ? or economic value was too low ?
  • Were tha large speies like sharks and turtles considered in statistics (even where released), their constant presence would demonstrate, that shrimp fishery does not harm the charismatic species

Reviewer 4 Report

Review

Paper title: Impact of trawling on the diversity of shallow water communities: A case study of Malindi-Ungwana bay, Kenya.

The authors conducted a long-term study based on fishery observational datasets to describe the current status of the shallow-water prawn trawl fishery in the Malindi-Ungwana Bay. The authors found an increase in both fishing efforts and catch over the period of 2016 to 2019.

They compared the compositions of retained and discarded catches and found significant differences with the species Pellona ditchela, Nematopalaemon tenuipes and Secutor insidiator being most responsible for the dissimilarity observed. The authors found similar species diversity and trophic level in the catch and concluded that these parameters were not affected by trawling. These results provide baseline data on the local prawn trawl fishery and may have important implications for the management regulations in Malindi-Ungwana bay.

All these reasons explain the relevance of the paper by Esther N. Fondo and co-authors submitted to "Diversity".

General scores.

The data presented by the authors are original and significant. The study is correctly designed and the authors used appropriate methods. In general, the statistical analyses are performed with good technical standards. The authors conducted careful work that may attract the attention of a wide range of specialists focused on fish biology and ecology.

Recommendations.

L 75. This section "In this study the catch…" should be a separate paragraph.

The author used a parametric test (2-way ANOVA). This approach requires normal distribution and heterogeneity of the data. The authors should provide the methods used to test these assumptions.

L 233. "p > 0.01". May be "p > 0.05"? "p > 0.01" does not mean a significant difference.

Fig. 7. The authors should increase the size of the small font.

Fig 8. Please, include information about the error bars.

Specific remarks.

L 9. Change “bottom dwelling” to “bottom-dwelling”

L 10. Change “amounts of by-catch is” to “amounts of by-catch are”

L 13. Change “twenty fold increase of” to “twenty-fold increase in”

L 17. Change “attributed” to “attributed to”

L 19. Change “four year” to “4-year”

L 28. Change “long term” to “long-term”

L 30. Change “generate” to “generates”

L 46. Change “contribute” to “contributes”

L 50. Change “by between 16 and 26 vessels landed between 400 and 1000  tons annually” to “landed between 400 and 1000  tons by 16–26 vessels annually”

L 65. Change “A management plan” to “The management plan”

L 69. Change “seasonal closer” to “seasonal bans”

L 70. Change “business plan” to “a business plan”

L 71. Change “use of” to “use of a”

L 72. Change “business plan” to “the business plan”

L 84. Change “operates” to “operate”

L 102. Change “northward flow” to “the northward flow”

L 105. Change “northward flowing East African Coastal Current meets the southward flowing” to “northward-flowing East African Coastal Current meets the southward-flowing”

L 111. Change “was used” to “were was used”

L 136. Change “position” to “positions”

L 139. Change “haul was” to “haul were”

L 140. Change “collected” to “were collected”

L 142. Change “obtain” to “obtained”

L 144. Change “catch composition estimation” to “the catch composition estimation”

L 163. Change “comparison” to “comparisons”

L 172. Change “total catch” to “the total catch”

L 187. Change “with increase” to “with an increase”

L 194. Change “catches” to “catch”

L 197. Change “1:9 during” to “1:9 from”

L 212. Change “all the four years” to “all years”

L 215. Change “all the four years” to “all years”

L 250. Change “Result” to “Results”

L 274, 280. Change “the years” to “across the years studied”

L 289. Change “small scale” to “small-scale”

L 289. Change “years that” to “years when”

L 292. Change “status” to “the status”

L 298, 299. Change “inconsistence” to “inconsistency”

L 300. Change “low level” to “a low level”

L 301. Change “observer data” to “observer dataset”

L 308. Change “landing” to “landings”

L 315. Change “small scale” to “small-scale”

L 317. Change “resulting” to “resulting in

L 340. Change “cost effective” to “cost-effective”

L 361. Change “small scale” to “small-scale”

L 379. Change “contributing” to “contributing to”

L 380. Change “species composition” to “the species composition”

L 403. Change “used is” to “used as”

Round 2

Reviewer 1 Report

My only request at this point, if possible, is that the bars in the histogram of Fig. 3 be placed side by side, as in Fig. 2. This makes the reading much easier.

Author Response

We have changed Figure 3 as requested by the Reviewer.

Thank you.